# Impact of Processing Parameters on the Quality of Pharmaceutical Solid Dosage Forms Produced by Fused Deposition Modeling (FDM)

**DOI:** 10.3390/pharmaceutics11120633

**Published:** 2019-11-27

**Authors:** Muqdad Alhijjaj, Jehad Nasereddin, Peter Belton, Sheng Qi

**Affiliations:** 1School of Pharmacy, University of East Anglia, Norwich, Norfolk NR4 7TJ, UK; muqdad.mousa@uobasrah.edu.iq (M.A.); J.Nasereddin@uea.ac.uk (J.N.); 2College of Pharmacy, University of Basrah, Basrah 61004, Iraq; 3School of Chemistry, University of East Anglia, Norwich, Norfolk NR4 7TJ, UK; P.Belton@uea.ac.uk

**Keywords:** Fused Deposition Modeling 3D Printing, processing parameters, pharmaceutical quality control, hot-melt extrusion, solid dosage forms

## Abstract

Fused deposition modeling (FDM) three-dimensional (3D) printing is being increasingly explored as a direct manufacturing method to product pharmaceutical solid dosage forms. Despite its many advantages as a pharmaceutical formulation tool, it remains restricted to proof-of-concept formulations. The optimization of the printing process in order to achieve adequate precision and printing quality remains to be investigated. Demonstrating a thorough understanding of the process parameters of FDM and their impact on the quality of printed dosage forms is undoubtedly necessary should FDM advance from a proof-of-concept stage to an adapted pharmaceutical manufacturing tool. This article describes the findings of an investigation into a number of critical process parameters of FDM and their impact on quantifiable, pharmaceutically-relevant measures of quality. Polycaprolactone, one of the few polymers which is both suitable for FDM and is a GRAS (generally regarded as safe) material, was used to print internally-exposed grids, allowing examination of both their macroscopic and microstructural reproducibility of FDM. Of the measured quality parameters, dimensional authenticity of the grids was found to poorly match the target dimensions. Weights of the grids were found to significantly vary upon altering printing speed. Printing temperature showed little effect on weight. Weight uniformity per batch was found to lie within acceptable pharmaceutical quality limits. Furthermore, we report observing a microstructural distortion relating to printing temperature which we dub The First Layer Effect (FLE). Principal Component Analysis (PCA) was used to study factor interactions and revealed, among others, the existence of an interaction between weight/dosing accuracy and dimensional authenticity dictating a compromise between the two quality parameters. The Summed Standard Deviation (SSD) is proposed as a method to extract the optimum printing parameters given all the perceived quality parameters and the necessary compromises among them.

## 1. Introduction

Three-dimensional (3D) printing continues to attract increasing attention in the pharmaceutical industry [1]. The flexibility and customisability potential offered by 3D printing promises to overcome some limitations of traditional pharmaceutical manufacturing techniques, particularly for personalised medicine [2]. The successful commercialisation of Spritam^®^ 3D printed levetiracetam tablets, which are produced by powder-jetting 3D printing, further cemented the prospects of 3D printing as a realistic technique for pharmaceutical manufacturing [3]. Several variants of 3D printing exist. Khaled et al. demonstrated the use of semi-solid extrusion 3D printing to fabricate 5-in-1 polypills [4]. Awad et al. used selective laser sintering to 3D print ibuprofen-paracetamol bilayered pellets [5]. Robles-Martinez et al. used stereolithographic 3D printing to print stratified polypills containing six drugs [6]. Fused Deposition Modelling (FDM) is another type of 3D printing variant that has been increasingly investigation for pharmaceutical exploration [1,7]. FDM is an inexpensive and very widely available technique compared to other variants of 3D printing. The pairing of FDM with Hot-Melt Extrusion (HME) [8,9,10] extends the library of available materials for 3D printing to include many pharmaceutically-relevant polymers, allowing for material flexibility when formulating a pharmaceutical product. Furthermore, FDM is not liable to some of the limitations posed by other 3D printing variants. Objects produced by powder-based 3D printing and semi-solid extrusion are susceptible to mechanical weakness issues [7,11]. Stereolithography relies on ultraviolet curing to fabricate the object, which raises drug stability and safety concerns [7]. Because of those reasons, an increasing number of publications suggests that FDM may be a favourable 3D printing method for the fabrication of personalized medicines in the form of solid dosage forms [8,11,12,13,14,15,16,17,18]. It was demonstrated that even off-the-shelf FDM printers are capable of producing proof-of-concept formulations.

There is currently no FDM printer that has been specifically designed for pharmaceutical applications. Commercially available printers, such as the MakerBot^®^ brand of printers, designed for fabricating plastic material-based prototype objects, have been frequently reported as the printers used when fabricating pharmaceutical dosage forms by FDM [1,8,9,14,15,16,18,19,20]. When characterizing the FDM printed objects, the non-pharmaceutical literature is largely concerned with the surface roughness and tensile strength of the fabricated objects [21,22,23,24,25]. In pharmaceutical applications, when the FDM printed objects are intended to be used as oral solid dosage forms, achieving precise and highly reproducible dimensions and weight is critical. However, to date, there is no debate on whether the fundamental design of these prototype-building FDM printers is suitable for pharmaceutical manufacturing. This study performed a detailed investigation of the influences of the processing parameters on the key quality attributes of the printed objects, including uniformity, reproducibility, and adherence to design specifications required for the pharmaceutical standards.

Although this study is limited to one particular printing machine it is intended to indicate the design parameters that must be considered when producing a 3D FDM printer for clinical pharmaceutical applications

In the FDM printing process, and any printing process, the quality of the printed object is the result of a complex interplay between the properties of the printing material and the settings of the machine (both adjustable and non-adjustable), the relevant parameters for FDM printing are shown in Figure 1. For the printer used in this work, the machine adjustable parameters are print speed, extrusion temperature, build plate temperature, and layer thickness. These represent the main variables in the FDM printing process. Temperature will necessarily affect the material properties, in particular, its rheological properties [26]. These, in turn, will determine the flow out of the printing head, often known as melt flow, which is one of the factors investigated in this study, and the spread on the build plate, which may then affect the reproducibility and dimensional precision of the printed object. Under some circumstances, high temperature may increase the fluidity of the printing material and cause unstable layer deposition. At the other extreme, low temperatures may cause reduced flow resulting in the print head, resulting in a blockage or a flow rate too small to print adequately at some printing speeds. Rheology may also affect flow after deposition on the build plate. In this work, we examine the interaction of the machine adjustable parameters and the print material rheological properties as indicated by the melt flow index and their effect on the reproducibility and adherence to design specification of the printed object. We note that some research suggests that computational factors such as slicing (the process by which the computer-generated 3D model is translated into movement instructions for the printer motors) algorithms could play a significant role in the overall quality of the printed object [27,28]. In this work, we restrict ourselves to exploring the relevant physical parameters, as other parameters cannot be manipulated without physically modifying the machine.

In addition, prior to the printing process, the ‘leveling’ of the printer, which is to ensure that the distance between the printing head and the build plate is constant across the printing area, should be performed to ensure the reproducible quality of the printed objects. It has been reported [8] that, for a machine of the same make and model, the outcome of this process could be operator dependent, indicating that the printing outcome may be irreproducible between operators. We report below on our examination of this problem.

Polycaprolactone (PCL), being one of the very few polymers approved for pharmaceutical use that is also suitable for FDM without further formulation [9,19], was used as the model polymer to investigate the effect of material and process parameters on the quality of printing. In the course of the work, it was noted that the first deposited layer tended to have different morphology to the layers above. We have dubbed this phenomenon The First Layer Effect (FLE). Apart from the variables discussed above, it is possible that the surface properties of the build plate may affect the first layer so the printing behaviour on different surfaces was also investigated. This is expected to have great importance for selecting the best build plate lining that allows the most suitable adherence of the printing material to the build plate during the building process and provides the easy peel-off of the object after the complete production of the printed dosage forms.

## 2. Materials and Methods

### 2.1. Materials

PCL commercial filament was purchased from MakerBot Industries (MakerBot Industries LLC., New York, NY, United States). PCL pure powder (CAPA™, Grade 6506) was purchased from Perstorp Chemicals (Perstorp Chemicals GmbH, Arnsberg, Germany), acetylsalicylic acid (ASA) was purchased from Sigma Aldrich (Sigma Aldrich Ltd., Dorset, United Kingdom).

### 2.2. Methods

#### 2.2.1. Preparation of Drug Loaded Filaments by HME

Three drug loaded PCL filaments containing 5%, 10%, and 15% ASA were prepared using HME. For each batch of the filament, the power mixes were accurately weighed then thoroughly mixed using a mortar and pestle. The powder mixes were fed into a Haake II minilab compounder (Thermo Scientific, Karlsruhe, Germany) equipped with a co-rotating twin screw set and a circular die with a diameter of 1.75 mm. Extrusion was performed at 100 °C with a screw rotation speed of 100 revolutions per second (RPM). All formulations were cycled in the extruder for 5 min to ensure homogeneous mixing of ASA and PCL prior to being flushed out.

#### 2.2.2. FDM 3D Printing of Commercial Filaments and Drug-Loaded Filaments

A MakerBot Replicator 2X twin nozzle desktop 3D printer (MakerBot Industries LLC, New York, NY, United States), equipped with nozzles of diameter 400 µm, was used for printing all filaments. The solid dosage form used in this study was a three-layered square film with 25 mm in length and width, and 0.6 mm in thickness (with 0.2 mm thickness per layer) (Figure 2). The film is comprised of 31 identical rods (rod dimensions: 0.4 mm wide, 0.2 mm thick, and 25 mm long). The design of the film is illustrated in Appendix A. The film was designed using Blender software and was then exported in stereolithography file format (STL file). The STL file was printed using MakerBot MakerWare™.

Twelve sets of different printing experiments were conducted, each varying either one of three factors (nozzle temperature, build plate temperature, and printing speed), as seen in Table 1. Five films were printed using each set of printing parameters, the printed film was weighed and the dimensions of the films were then accurately measured using a digital calliper. Printing of the drug-loaded filaments was performed using the MakerBot Replicator 2X printer. The filaments were printed with a nozzle temperature of 100 °C, an unheated build plate and were printed at a speed of 90 mm/s.

#### 2.2.3. In Vitro Drug Release Studies

ASA release studies from the 5%, 10%, and 15% loaded 3D printed grid films were conducted using the United States Pharmacopoeia (USP) apparatus II (paddle) using a revolution rate of 50 RPM. The grids were placed in 750 mL of pH 1.2 simulated gastric fluid for 2 h, then transferred to 900 mL of pH 6.8 phosphate buffer saline. Determination of amount of ASA released was conducted using a Perkin Elmer Lambda 35 UV/Vis spectrophotometer (PerkinElmer, Inc., Waltham, MA, United States) at a wavelength of 265 nm [29].

#### 2.2.4. Melt Flow Index Measurements

The melt flow index (MFI) of the filaments was measured using an adaptation of the ISO 1133-1 standard method [30]. The printing head of a MakerBot^®^ Replicator 2X desktop FDM 3D printer (MakerBot Industries LLC. New York, NY, United States) was detached from its roller feeding zone and used to melt the filaments. A piston attached to a metal weight of 80 g was used to propel 2 cm slices of the filament through the nozzle (Appendix A). The filament melt flowing from the nozzle was collected at specified time intervals and weighed. The mass of the polymer (in milligrams) extruded per second was regarded as the MFI of the polymer. Using this adapted method maintains the gravimetric flow of the polymer melt that is characteristic of standard MFI measurements, but over the relevant capillary diameter of the printing nozzle. The MFI measurements for PCL were performed using a temperature range of 70 °C to 130 °C.

The quantity (in milligrams) of material deposited per second during an actual FDM printing process is defined in this study as the FDM-MFI. The FDM-MFI values of the PCL filaments were measured at three different printing speeds (30 mm/s, 90 mm/s, and 160 mm/s) over a range of processing temperatures (70–130 °C). For both MFI and FDM-MFI measurements, the rate of deposition of the objects was then expressed as weight deposited per unit time (mg/min).

#### 2.2.5. Leveling of the Build Plate of the Printer

Two investigations of the platform leveling were performed. Inter-person calibration: the platform was leveled by two operators, then six square films were printed and their weights and dimensions quantified. Same person inter-day calibration: the platform was leveled by the same operator on two different days; then, a set of six squares were printed and their weights and dimensions were quantified. The object used for calibration was a simple square design with an edge length of 10 mm and a thickness of 1 mm printed using standard printer settings at 100 °C.

Leveling was conducted as per the standard printer leveling procedure outlined in the user manual of the printer [31]. The card supplied with the instrument was placed between the nozzle and the build plate at various positions around the build plate and the leveling screws were adjusted until the card can just slide between the build plate and the nozzle.

#### 2.2.6. Printing on Different Surfaces

The FLE was further investigated by printing the PCL films on three different surfaces, Kapton^®^ (polyimide) Tape, aluminum, and glass. Aluminum and glass were secured to the build plate using double adhesive tape, to ensure that there was no movement of the test surface during printing. All of these printing experiments were conducted using a nozzle temperature of 100 °C and a printing speed of 90 mm/s. The build plate was leveled following the protocol described in 2.2.4 before each printing attempt.

#### 2.2.7. Characterization of Printed Solid Dosage Forms

Microscopic images were acquired using a Linkam Imaging Station, equipped with a Linkam MDS600 heating/cooling stage (Linkam Scientific Instruments, Tadworth, United Kingdom). Image analysis (dimensions measurements) was conducted using LINK software version 1.0.5.9 (Linkam Scientific Instruments, Tadworth, United Kingdom) to measure the road widths of the 3D printed objects.

Fourier-Transform Infrared (FTIR) spectroscopy was used to characterize all materials used in this study. Spectra were acquired using a Vertex 70 infrared spectrometer (Bruker Optics Ltd., Coventry, United Kingdom), equipped with Golden Gate, heat-enabled Attenuated Total Reflectance (ATR) accessory (Specac Ltd., Orpington, United Kingdom) fitted with a diamond internal reflection element. The spectra were acquired in absorbance mode, with a resolution of 2 cm^−1^, and 32 scans per sample. Scanning range was set to 4000 cm^−1^–600 cm^−1^. Spectral analysis was conducted using OPUS software, version 7.8 (Bruker Optics Ltd., Coventry, United Kingdom).

Differential Scanning Calorimetry (DSC) was carried out using a TA Universal Q2500 Discovery series DSC (TA Instruments, Newcastle, DE, United States). DSC was used to characterize the pure materials (PCL commercial filament, PCL powder, and ASA powder), the physical mixes, and extruded filaments. PCL powder was characterized using a heat-cool-reheat method with a range from 20 °C to 120 °C, followed by cooling to −90 °C, then reheating to 120 °C. ASA powder and physical mixes of all three formulations were scanned using a heat-cool-reheat method; samples were tested over a range of 20 °C to 150 °C, then cooled to −90 °C, and then reheated to 150 °C. All extruded filaments were characterized using a heat-cool-reheat method using a temperature range of −90 °C to 150 °C. A heating/cooling rate of 10 °C/min was used for all experiments. All samples were equilibrated for 3 min at either 20 °C or −90 °C at the start of each experiment. All characterization experiments were carried out in triplicates.

#### 2.2.8. Statistical Analysis

Independent sample T-tests were conducted using Microsoft Excel (version 2016) expanded with the statistical data analysis add-on.

Summed Standard Deviation (SSD) was calculated as follows: for each value of temperature (T) and speed (S) the normalised standard deviations ($P^{\prime })$ of the measured parameters weight (M), length (L), width (W), thickness (D), and road width (R), were calculated using the following equation:(1)$$P_{i}^{\prime }=(\frac{\sigma_{i}^{p}}{P_{i}}{)}_{T,S}$$
where $P_{i}$ is the value of the measured parameter at a fixed value of S and T and $\sigma_{i}^{p}$ is its standard deviation. These were summed up, as shown in Equation (2):(2)$$SSD=\sum_{\begin{array}{c} S=0\\ T=0 \end{array}}^{\begin{array}{c} S=n\\ T=n \end{array}}{\left[{P^{\prime }}_{M}+{P^{\prime }}_{L}+{P^{\prime }}_{W}+{P^{\prime }}_{D}+{P^{\prime }}_{R}\right]}_{T,S}$$

Principal Component Analysis (PCA) was conducted on all the measured responses using IBM^®^ SPSS statistics (version 25). Following the Kaiser criterion, only principal components with an eigenvalue ≥ 1 were extracted (*n* = 2), yielding a total explained variance of 82.15% (Principal Component 1: 57.77%. Principal Component 2: 24.38%). The unrotated components matrix showed no factor that could be explained solely by either principal component (coefficient < 0.4); therefore, Varimax rotation was conducted such that each factor is solely described by a single principal component. The loadings plot was used to extract variable scores for each of the two factors (printing temperature, and printing speed), the scores were imported into Microsoft Excel and used to generate a biplot [32,33].

## 3. Results and Discussion

### 3.1. Impacts of Build Plate Leveling

It has been previously reported in the literature that the leveling of the build plate of MakerBot^®^ printers causes a significant difference in the weights of the 3D printed objects when conducted by different operators [8]. In an attempt to prevent this discrepancy from misattributing other findings of this work, calibration prints were conducted as described in Section 2.2.5. The weights and dimensions of the two sets were then compared via an Independent Sample T-Test. The results are shown below in Table 2 and Table 3.

As seen in Table 2, the *P*-value for the *T*-tests for all of the parameters was >0.05, indicating no significant difference between the objects printed when the leveling was carried out by two different operators.

Table 3 shows the results of the calibration prints when the platform was leveled by the same operator on two different days. The P-value for width was the only T-Test parameter found >0.05, indicating no significant difference. P-values for length, thickness, and weight were all <0.05, indicating significant difference.

A study conducted by Melocchi et al. using the same model of the FDM printer [8] previously reported a significant difference when the build plate was leveled by different operators. We acknowledge that the results reported herein contradict with those findings. However, as shown in Table 3, we observed a significant difference (most importantly in weights of the printed objects) when the printer was leveled by the same operator on two different days. Both our findings and what was reported by Melocchi et al. serve to highlight the operator dependence of the calibration of this model of printer which can be a potential problem in this design of the printer if it is used for printing pharmaceutical standard products. The current method employed by the printer to level the build plate is very subjective, relying on what the operator’s judgment as ‘suitable friction’ between the nozzle tip and leveling card. While this may be adequate for printing large commercial prototype objects, it is unlikely to be a suitable ‘calibration’ method for pharmaceutical printing of oral solid dosage forms where high printing precision is required. Therefore, we believe that the leveling process of any printer to be used for manufacturing pharmaceutical standard solid dosage forms needs to be modified to a more robust procedure.

### 3.2. Impacts of Melt Flow of the Printed Materials

The Melt Flow Index (MFI), is defined by the ISO standard 1133-1 as “the mass of the molten polymer, in grams, that flows through a capillary of a specific diameter in 10 min” [30]. The MFI of a filament is often cited as one of the key factors defining the success of an FDM printing process [9,19,34]. The MFI is highly associated with the thermal viscosity of the filament materials at a certain printing temperature [26]. Therefore, MFI measurements (which describe the amount of PCL deposited due to its melt flow) were compared to the amount deposited during FDM printing (referred to as FDM-MFI in Figure 3b). Despite the temperature dependence of rheology seen in the MFI experiments (Figure 3a), there was a very weak temperature dependence of deposition rates when printed using FDM (Figure 3b). This implies that over the range of relevant printing temperatures, the change in the melt flow of the filament is not too great to significantly impact the amount of material being deposited by the print head. The only effect was seen at 70 °C, where printing was only possible at a printing speed of 30 mm/s. This leads to the hypothesis that, provided that the operating speed is sufficient to allow free flow of the polymer melt, the contribution of operating temperature to the printability of the material is not of prime importance.

### 3.3. Impact of Processing Conditions on Weight Uniformity

MFI measurements showed that PCL has measurable melt flow index at a temperature of 70 °C; however, printing of PCL was only possible at this temperature at the lowest speed of 30 mm/s. In addition, objects printed at 80 °C and either 90 mm/s or 160 mm/s were much distorted, with very poor printing quality and erratic melt deposition. Therefore, samples printed at 70 °C and 80 °C were disregarded. This suggests that there is a lower limit to the melt flow index, below which good quality printing is not possible.

Despite that, on average, the impact of the printing temperature on melt deposition rate was not as significant as printing speed (Figure 3b). It appears that it may impact the reproducibility of the weights of the printed objects. As can be seen in Figure 4a, at a fixed printing speed, increasing the printing temperature resulted in an increase in the weights of the printed objects. This is to be expected, as melt flow of PCL increases with temperature. The standard deviations of weights do not seem to follow any trends either with temperature or printing speed. The largest recorded standard deviation was ±3.2 mg at 90 mm/s and 120 °C. The second largest being ±3.0 mg at 30 mm/s and 70 °C, and the third largest being ±2.4 mg at 30 mm/s and 110 °C. Notably, fixing the printing speed at 160 mm/s yields the narrowest standard deviations, with the smallest standard deviation at that speed being ±0.1 mg at 100 °C, and the largest being ±0.9 mg, seen at both 120 °C and 130 °C. Even though all the reported standard deviations fall well within acceptable limits for weight uniformity specified in the pharmacopoeias, it is worth noting that changing the printing conditions was seen to substantially impact weights of the printed objects, by as much as 31.3 mg. The lightest printed object weighed 169.1 mg (±0.5 mg, printed at 70 °C, and 160 mm/s), and the heaviest printed object weighing 200.3 mg (±1.5 mg, printed at 130 °C, and 30 mm/s). These results indicate that if FDM is to be utilized in personalized medicine, careful screening of the printing parameters, and choosing the appropriate conditions to match the target dose is of utmost importance [2].

### 3.4. Impact of Processing Conditions on Dimensional Authenticity

For the commercial printers, the pre-set (target) object parameters are dimensions instead of weight. Therefore, it is important to understand the effect of the process parameters by comparing the measured printed object dimensions to the target values pre-set by the STL file. Despite the source STL file being designed as a square in this study, all the objects printed displayed a difference between length and width. To avoid ambiguity, length was defined as the dimension parallel to the roads of the first layer, and width as the dimension perpendicular to the roads of the first layer. The impacts of the printing temperature on the dimensions (width, length, and thickness) of the objects are illustrated in Figure 4b,c. For each printing condition, the length of the objects was found to be larger than the width of the objects. The printing temperature shows no significant effect on the reproducibility of the length and width of the printed films at a fixed printing speed.

Reducing the printing temperature increases the thickness of the films for both 30 and 160 mm/s printing speeds but there is no significant effect at the printing speed of 90 mm/s. Microscopic imaging of the printed films (Appendix A) revealed that the road width of the first layer is much larger than that of the subsequent layers, and the width of the first layer appears to be correlated with the nozzle temperature. This correlation between first layer road width and the reduction of object thickness with increasing temperature suggests that when the first layer is deposited on the build plate, the fluid melt spreads sideways, increasing in width and decreasing in thickness, resulting in the observed effect on road width and thickness. We have chosen to dub this phenomenon the First Layer Effect (FLE), which is discussed in a further section.

Printing at 90 mm/s and 160 mm/s yielded objects that possessed greater length and lower width than the target (25.29 mm ± 0.06 mm × 24.95 mm ± 0.04 mm for 90 mm/s, and 25.29 mm ± 0.11 mm × 24.93 mm ± 0.04 mm for 160 mm/s), showing no significant difference in dimensions between the two conditions. However, printing at 30 mm/s yielded objects that were smaller than the target geometry of 25 mm × 25 mm (with the films being 24.68 mm ± 0.06 mm × 24.33 mm ± 0.11 mm). At this speed, no changes in length and width relative to changing the temperature were observed. In terms of reproducibility of lengths and widths, none of the three printing speeds showed remarkably different results, with no significant differences between the standard deviations of dimensions between the three printed conditions.

As can be seen in Figure 4c, the thickness of the 3D printed objects tends to decrease with increasing temperature for both 160 mm/s and 30 mm/s, going from 0.65 mm ± 0.00 mm at 90 °C to 0.61 mm ± 0.01 mm at 130 °C when printed at 30 mm/s, and from 0.65 mm ± 0.02 mm to 0.59 ± 0.01 mm when printing at 160 mm/s. Objects printed at 90 mm/s had, on average, consistent thickness, independent of printing temperature. Furthermore, printing at 90 mm/s appears to have the most reproducible object thicknesses, with the narrowest recorded standard deviation at said speed being ± 0.00 mm, and the widest being ± 0.01 mm (within the limits of detection of the digital caliper used).

### 3.5. The First Layer Effect (FLE)

Figure 5a shows the average road width of each layer under different printing conditions. The top layer is the closest to the nozzle. The platform layer is the bottommost layer resting on the build plate of the printer. No significant difference was seen between the top and middle layers of the objects, regardless of printing conditions. The average road width for the top layer was 347.0 µm ± 12.00 µm at 30 mm/s, 345.0 µm ± 15.37 µm at 90 mm/s, and 323.5 µm ± 25.62 µm at 160 mm/s. The average road width for the middle layer was 341.3 µm ± 18.46 µm at 30 mm/s, 310.4 µm ± 13.56 µm at 90 mm/s, 316.8 µm ± 27.45 µm at 160 mm/s. The platform layer displayed a larger road width than the corresponding top and middle layers at every printing condition, with the average road width of the platform layer being 438.8 µm ± 51.79 µm at 30 mm/s, 425.8 µm ± 34.56 µm at 90 mm/s, 436.1 µm ± 68.63 µm at 160 mm/s.

The road width of the platform layer was also observed to vary inversely with printing temperature. There was a notable decrease in the average road width of the first layer at different printing temperatures; at 30 mm/s, the road width increased from 402.1 µm ± 12.9 µm at 90 °C to 517.5 µm ± 26.0 µm at 130 °C. At 90 mm/s, the road width increased from 420.5 µm ± 19.1 µm at 90 °C to 465.2 µm ± 14.9 µm. At 160 mm/s, the road width increased from 422.0 µm ± 35.7 µm at 90 °C to 554.0 µm ± 55.9 µm at 130 °C. Due to the platform layer being the first layer constructed during the fabrication of the object, this phenomenon has been dubbed The First Layer Effect (FLE). This spreading effect is assumed to be caused by the nature of the interaction of PCL with the surface of the Kapton^®^ tape. Therefore, printing was attempted on different surfaces. Printing on different surfaces yielded different spreading amounts (Figure 5b). Building on glass was found to yield the narrowest average road width (322.3 µm ± 13.03 µm). Building on aluminum was found to yield the widest road width (419.8 µm ± 22.00 µm). The increased spreading with temperature when printing on Kapton^®^ is likely due to decreased viscosity of PCL at higher temperatures allowing it to flow more prior to solidifying.

When printed at higher temperatures (>110 °C), PCL was found to bind strongly to the Kapton^®^ tape. When printed at lower temperatures (i.e., 80 °C–100 °C), the PCL films were easily removed. However, no sticking to aluminum or glass was observed. The temperature-dependent spreading, and the sticking of the objects to the Kapton^®^ tape at elevated temperatures is probably due to the formation of PCL-polyimide interactions at elevated temperatures, yielding greater wetting and adhesion to the surface.

### 3.6. Impact of Drug Incorporation

Drug incorporation can often cause changes to the printability of polymeric filaments, in most cases due to the plasticisation effect of the drug when a molecular dispersion is formed between the drug and the polymer. Three levels of drug loading were used to create filaments that were either true molecular dispersions or supersaturated with the crystalline drug in the filaments. DSC and ATR-FTIR spectroscopy were used to characterize the physical state of the ASA in the filaments prepared by HME (Appendix A and Figure 6). The results indicate the formation of a molecular dispersion of ASA in PCL at 5% drug loading, whereas both the 10% and 15% formulation contain a crystalline fraction of ASA. The shifted peaks of the C–O and C–OH groups indicate the molecular interaction of ASA and PCL via hydrogen bonding at the carboxyl groups, as seen in Figure 6. The detailed analysis of the formulation characterisation data can be found in the Appendix A.

It would be expected that a drug-polymer molten solution would exhibit different spreading behaviour when printed on the surface since the incorporation of the drug would be expected to alter the physical properties of the mix [35]. This was investigated by printing ASA-loaded PCL filaments at a median condition of 100 °C and 90 mm/s. Followed by determination of the average road width per layer. The results can be seen in Figure 7 against a placebo filament printed at the same conditions. No significant difference was seen in the nozzle and middle layer between the three drug-loaded filaments and the placebo filament. However, there is a significant increase in the average road width of the first layer that was brought about by incorporation of ASA in PCL; printing at 90 mm/s and 100 °C showed an increase in the first layer road width from ≈390 µm to >600 µm. This is likely due to the presence of ASA in the PCL matrix decreasing the viscosity of the melt, allowing for a greater extent of spreading before the road completely solidifies. No significant difference was seen between the three drug-loaded formulations, this could be attributed to the ASA-PCL melt reaching the maximum possible wettability it can achieve on Kapton^®^ before it solidifies, regardless of drug loading. Notably, the drug-loaded objects exhibited greater sticking to the platform than their placebo counterparts, requiring very careful peeling off the platform with a razor blade to avoid severely deforming the object printed.

### 3.7. In Vitro Drug Release Studies

Figure 8 shows the in vitro drug release rates for the 5%, 10%, and 15% ASA-loaded formulations in PCL. A significant difference in the release rate of the 5% formulation compared to its higher loading counterparts was observed. Both the 10% and 15% formulation achieved ~100% release in under 300 min. The 5% drug-loaded formulation, however, had released only ~30% of the drug after 8 h. As previously discussed, the 5% drug-loaded formulation was the only formulation in which the ASA was molecularly dispersed within the matrix of PCL, while both 10% and 15% drug-loaded contained phase separated crystalline ASA. This is further evidenced by the aforementioned differences in release rate. PCL is a biodegradable polymer that is commonly used for implantable, long-term release formulations [36]. It is insoluble in aqueous media, and only degrades over time via hydrolysis of its ester linkages in physiological conditions. In the 5% drug-loaded formulation, the ASA is molecularly dispersed within the polymer, thus the drug release relies on the slow diffusion of ASA molecules from the PCL matrices and the polymer degradation. On the other hand, the 10% and 15% drug-loaded formulations contain phase–separated crystalline ASA which can dissolve much faster. No visible disintegration of the FDM printed films was observed over the 8 h dissolution tests.

### 3.8. Statistical Analysis

The data above demonstrated the quality of the printed dosage form is a result of the complex interplay between different processing and materials factors. These factors also often interact such that varying the level of factors concurrently has a greater impact over a perceived measure of goodness than varying either parameter individually. Furthermore, while one can measure particular properties of the 3D printed object (such as weight, dimensions, road width, etc.) to be utilized as measures of goodness for parameter selection, this remains a non-straightforward process. Mainly because processing conditions that appear to produce a more favoured object when observing one measure of goodness fail when another measure of goodness is considered (i.e., weight versus dimensional authenticity). Therefore, it is clear that there exists a need for an overarching method for selecting the optimum printing conditions that will produce the objects with the greatest overall quality. For this purpose, we propose basing this method on a measure of goodness to determine the printing conditions which will produce objects with the highest overall printing reproducibility. Said conditions are those that will yield the minimum Summed Standard Deviation (SSD) score. This proposed SSD can be calculated by summing the standard deviations of each measured value at each condition.

The SSD scores for all printing conditions can be seen in Figure 9. The processing parameters 120 °C and 90 mm/s yielded the lowest SSD while 130 °C and 160 mm/s yielded the highest SSD. Objects printed at 90 mm/s notably had lower SSD scores for every single temperature than their counterparts printed at 30 mm/s and 160 mm/s. The SSD scores represent a figure of merit, which can be used to select a set of printing conditions which will give minimal overall variability.

While the SSD provides a quick method to determine the optimum printing conditions for a given filament, further statistical analysis techniques can be used to extract more information about the process parameters and how they interact to influence the process. Therefore, PCA was conducted as an exploratory data analysis tool to investigate the interplay between the different perceived quality parameters.

Figure 10 shows the loadings plot of the measured responses in rotated space. Principal Component 1 (PC1) was found to describe object mass, road width, length, width, and the printer deposition index, corresponding to 57.77% of the total variance. Principal Component 2 (PC2) was found to describe object thickness, and first layer width, corresponding to 24.38% of the total variance.

As discussed in previous sections, object thickness and the first layer effect were found to correlate more strongly with printing temperature than with printing speed (Figure 4c and Figure 5a), while object mass, length and width, and the FDM-MFI were found to vary more significantly in response to change in printing speed rather than printing temperature. Therefore, one may extrapolate that PC1 may be redubbed the speed axis, as it describes variance introduced due to change in printing speed. Similarly, PC2 may be named the temperature axis as the variables it describes are those that alter more significantly in response to changes in printing temperature. Since PC1 accounts for the majority of the total explained variance (57.77%), one may deduce that printing speed a more significant contributor to the perceived quality parameters of FDM printed objects than printing temperature.

Observing the measured variables as described by the loadings plot allows for a more overarching look at how printing speed and temperature both influence perceived quality attributes, as well as how the quality attributes relate to one another. Object thickness and first layer width load opposite to each other on the temperature axis, suggesting the two are anti-correlated. This suggests that the spreading of the first layer not only increases its road width, but also decreases road height in the Z-axis. Furthermore, first layer width loads positively, while thickness loads negatively on the axis, indicating that wider road widths are brought about by higher temperatures, while greater object thickness is a result of lower printing temperature. Therefore, a conclusion can be drawn that printing at higher temperatures leads to a more drastic FLE, while printing at lower temperatures leads to thicker objects.

Length, width, mass, FDM-MFI, and road width were all described by the speed axis. Length, width, and FDM-MFI being anti-correlated to mass and road width, with the former three loading positively, and the latter negatively, indicating that length, width, and the FDM-MFI are directly correlated to printing speed, while mass and road width are inversely correlated. The length/width versus road width correlation is an interesting one as it gives insight into the operation of the feeding motors, as well as printer accuracy; the anti-correlation between road width and printing speed suggests that at higher printing speeds, the printer is not feeding sufficient material to keep up with the demands of the higher printing speed, leading to the road being deposited to be tugged as the print head is moving, stretching it thinner (leading to a decrease in road width) and longer (leading to an increase in length/width). This argument relating printing speed and feeding speed suggests that, at higher print speeds, the printer is not providing enough material feed to faithfully replicate a print at lower speeds. Therefore, objects printed at higher speeds should have less mass than their lower speed counterparts. This was found to be true as mass loaded negatively on the speed axis, and was found to be anti-correlated to length, width, and FDM-MFI. The latter, which was found to increase relative to speed (Figure 3b) was, unsurprisingly, found to load positively on the speed axis.

Figure 11 shows the biplot obtained when case scores were projected onto the loadings plot shown in Figure 10. The X-axis, denoting the scores of the cases against PC 1, unsurprisingly separates the cases into three clusters relative to printing speed, with three clusters showing clear separation between the 30 mm/s set, followed by 90 mm/s, and followed by 160 mm/s. The clustering pattern fits the argument presented prior relating printing speed to object mass and dimensions, as the leftmost cluster, falling on the “largest mass” quadrant of the biplot belonged to the 30 mm/s, then 90 mm/s, which was then followed by 160 mm/s, the “widest dimensions” set. Notably, there is less separation between the latter two sets than between 30 mm/s and 90 mm/s, which strongly mirrors the FDM-MFI results displayed in Figure 3b.

The Y-axis, which shows the scores loadings relative to PC2 (the temperature axis) describes quality parameters which are influenced by printing temperature (object thickness, and the FLE). While the clusters do not appear to offer any meaningful metric towards either parameter at first glance, closer observation reveals that both object thickness and FLE are not described within each cluster, but rather between the clusters. With respect to object thickness, looking at objects printed at the same temperature (i.e., 90 °C), the one printed at 160 mm/s was the thickest, followed by the object printed at 30 mm/s, with 90 mm/s coming in third. Similarly with respect to FLE, of the objects printed at 100 °C, the one printed at 160 mm/s had the largest FLE, followed by the one printed at 30 mm/s, followed by the one printed at 90 mm/s. This inter-cluster pattern was found to apply to all the observed cases. This data pattern suggests that there may be either a more complex physical phenomenon, or a speed–temperature interaction that skews what is perceived as the impact printing temperature has on the quality parameters.

Of the three clusters, the 90 mm/s group appears to show the least variance relative to change in printing temperature. This indicates that that printing speed further minimizes the significance of printing temperature. Furthermore, the 90 mm/s cluster was the most centered cluster with respect to the four quadrants of the biplot, indicating that 90 mm/s offers the best compromise between the opposing quality parameters. This sits in agreement with the result obtained from the SSD displayed in Figure 9, in which printing at a speed of 90 mm/s yielded the lowest SSD, regardless of printing temperature, indicating that higher reproducibility is achieved when printing at 90 mm/s. Therefore, one may extrapolate that, for the PCL filament used herein, printing at 90 mm/s offers the most predictable and reproducible results, making it the optimum printing speed for this filament.

## 4. Conclusions

The results of this study demonstrated the significant impact processing parameters have on some of the perceived quality attributes of 3D printed dosage forms (weight, dimensional authenticity, road width, and overall print reproducibility). For the printer used in the study, printing speed exhibited a more profound impact on the weight uniformity and dimension authenticity of the printed dosage forms than printing temperature. Printing temperature and the build plate surfaces were found to contribute significantly to the FLE.

The repeatability and consistency of the calibration of the print was examined using the statistic variation observed in build plate leveling. The results confirmed that the build plate leveling is a source of significant operator-dependent error. This finding indicates the needs to redesign the build plate leveling mechanism to allow more consistent and operator-independent calibration of the printer.

The use of summed standard deviations (SSD) enables the calculation of a figure of merit indicating the most reproducible set of printing conditions, and by extension, the optimal printing parameters.

PCA was used as factor exploration method to further investigate the impact of processing parameters on the quality attributed measured herein, and possibly explore some factor interplay that may not necessarily be obvious by looking at the impact processing parameters may have on a single quality parameter outside the context of the entire set of quality attributes.

By using PCA, printing speed was found to explain more data variance than printing temperature. Factoring in that printing was not possible at a printing temperature of 70 °C and printing speed ≥90 mm/s, we can conclude that, provided that the temperature is sufficient to overcome polymer viscosity, printing speed will have a greater contribution to the perceived quality parameters of the printed object.

Printing speed was found to be negatively correlated with object road width, and more notably with object mass. Control of the latter being critical for accurate dosing in a pharmaceutical manufacturing context. Which leads us to deduce that increasing printing speed does not equal increased printing throughput, and is not a valid method for the scaling-up of a pharmaceutical FDM printing process.

The use of PCA did prove to be an effective chemometric method to determine the optimum printing conditions for the filament studied herein, as it revealed the processing parameters that produce objects that offer the best compromise between the extremes of all the perceived quality parameters. Said processing conditions were deemed to be the optimum because they offer the most reproducible and predictable prints, and not because they match a desired target value.

For pharmaceutical applications, the control of such impacts should be thoroughly understood as it can affect the performance of the printed formulation. These results brought to the conclusion of that careful engineering for a pharmaceutically suitable FDM 3D printer should be treated as a priority for making the shift of FDM 3D printing from proof-of-concept to industrial application.

## Figures and Tables

**Figure 1 pharmaceutics-11-00633-f001:**
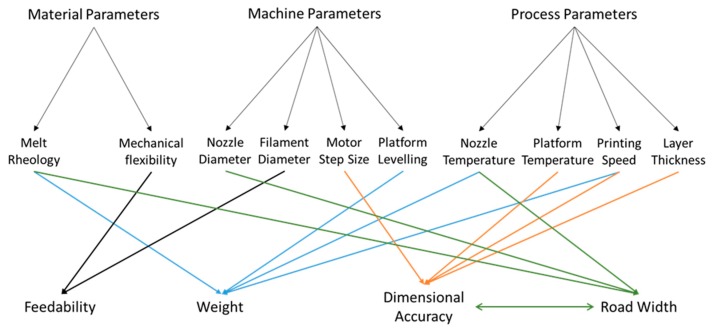
Summary of the interactions between the materials properties and the machine and process parameters in a fused deposition modeling (FDM) three-dimensional (3D) printing process.

**Figure 2 pharmaceutics-11-00633-f002:**
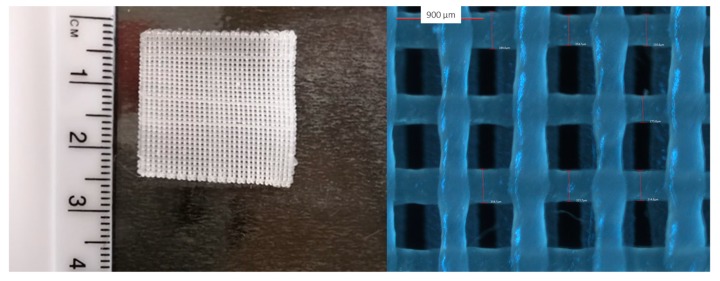
Macroscopic (left) and light microscopy image (right) of the 3D printed film.

**Figure 3 pharmaceutics-11-00633-f003:**
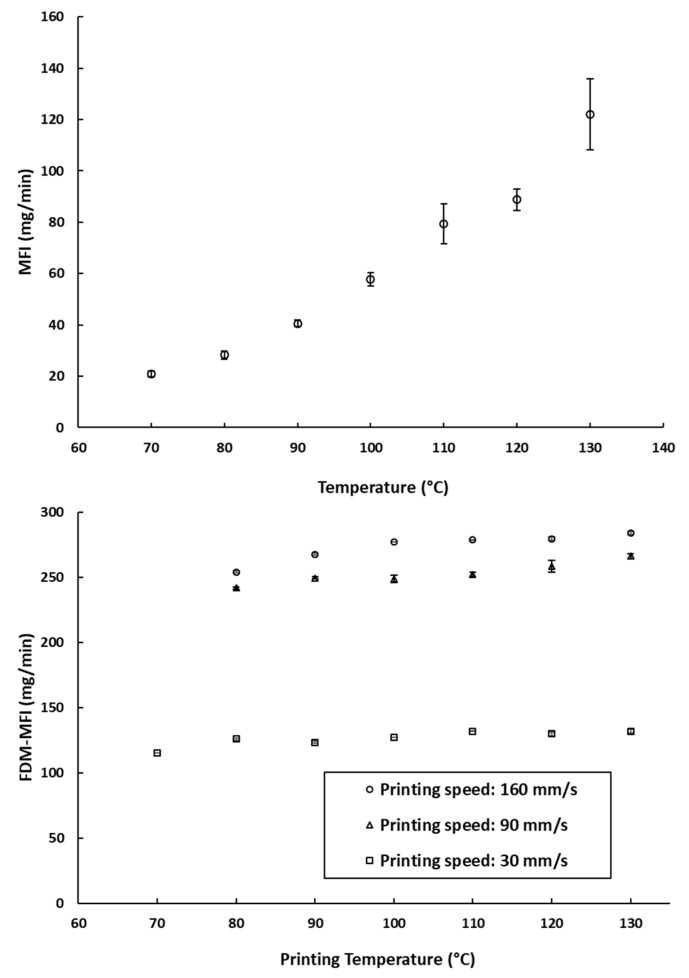
(**a**) The measured MFI values of the PCL filaments at different temperatures (the arrow indicates the manufacturer recommended printing temperature); (**b**) FDM-MFI of the PCL filaments measured at different printing speeds across different printing temperature. *: printing at 90 mm/s and 160 mm/s was not possible at this speed.

**Figure 4 pharmaceutics-11-00633-f004:**
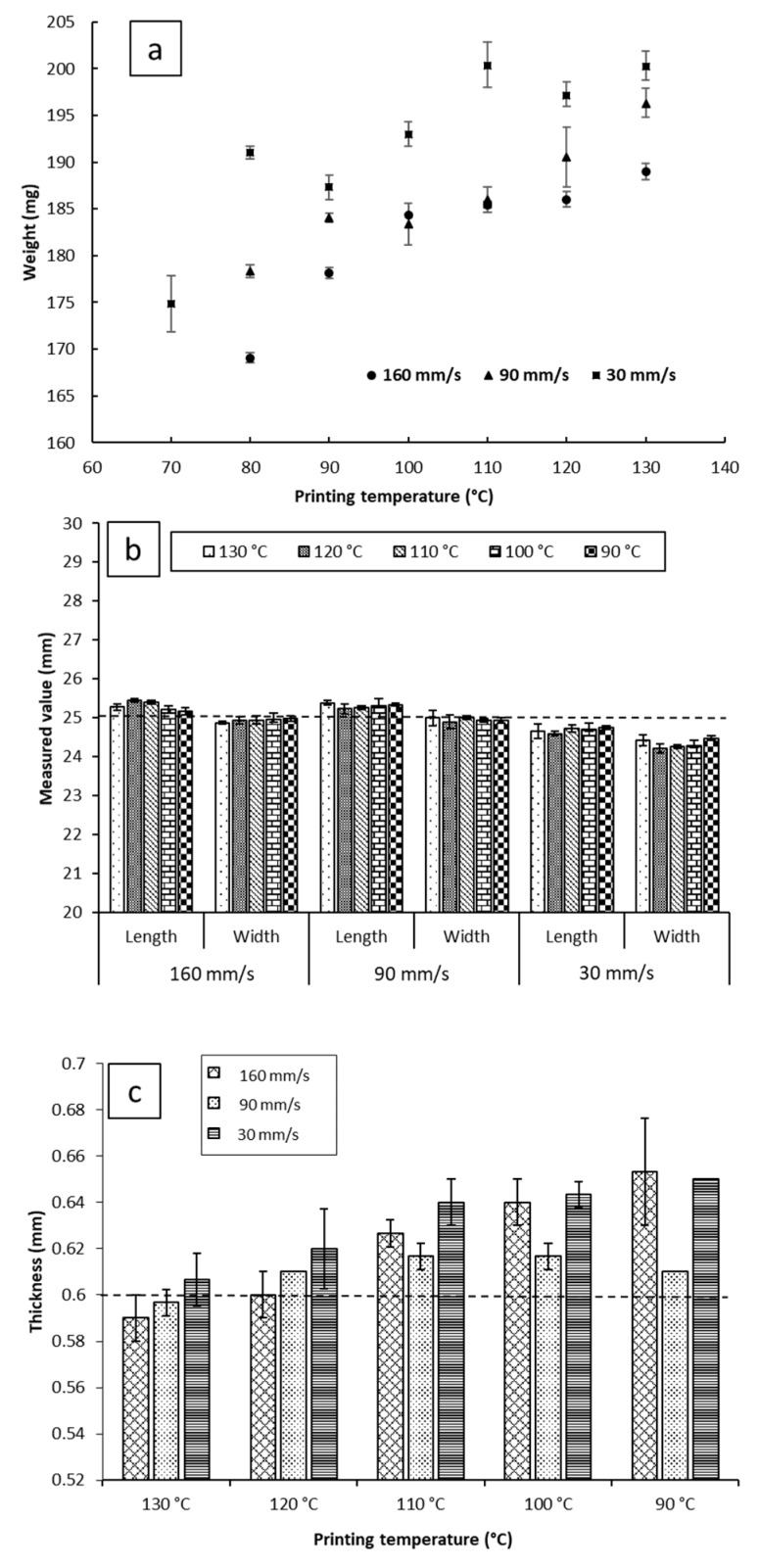
(**a**) The weight; (**b**) the length and width; and (**c**) the thickness of the films printed at a range of speed and temperature. The dotted lines at 25 mm in (**b**) and 0.6 mm in (**c**) indicate the theoretical targeted values of the dimensions.

**Figure 5 pharmaceutics-11-00633-f005:**
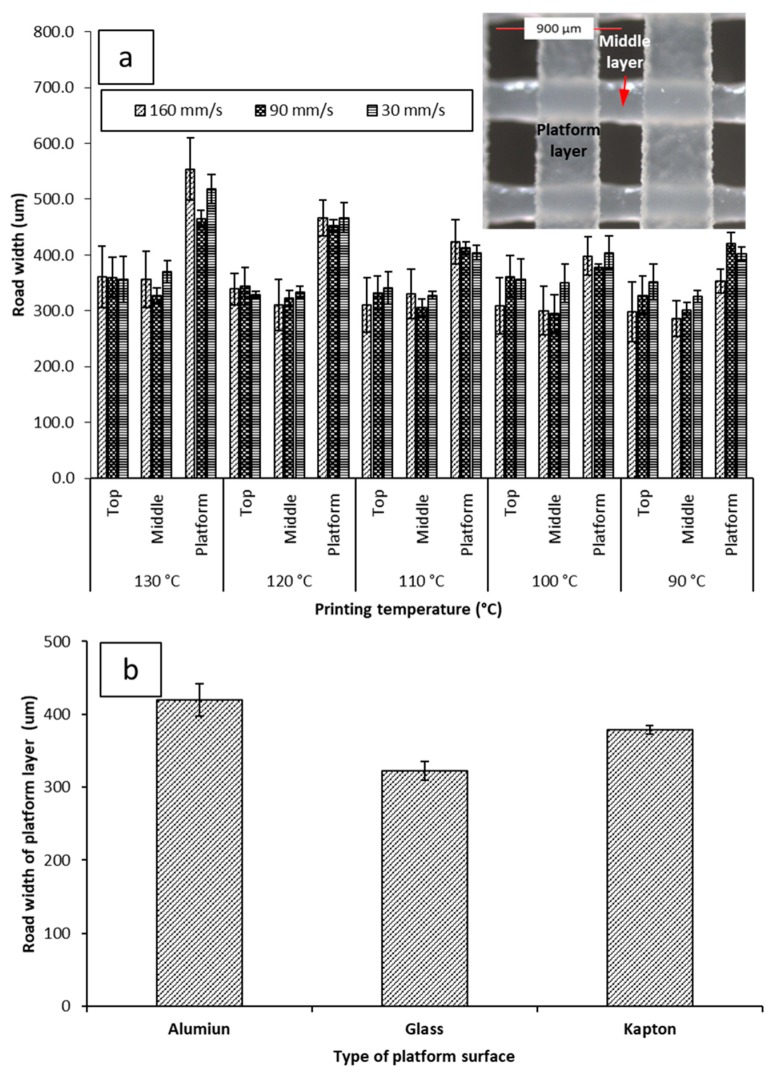
(**a**) Impacts of printing temperature and printing speed on the road width measurements of the platform (the first layer deposited on the build plate), middle, and top layer (with inserted microscopic image illustrating the platform and mid layers); (**b**) impact of different printing surfaces on the road width of the platform layer when printed at 90 mm/s and 100 °C. Microscopic images of printed films can be seen in Appendix A.

**Figure 6 pharmaceutics-11-00633-f006:**
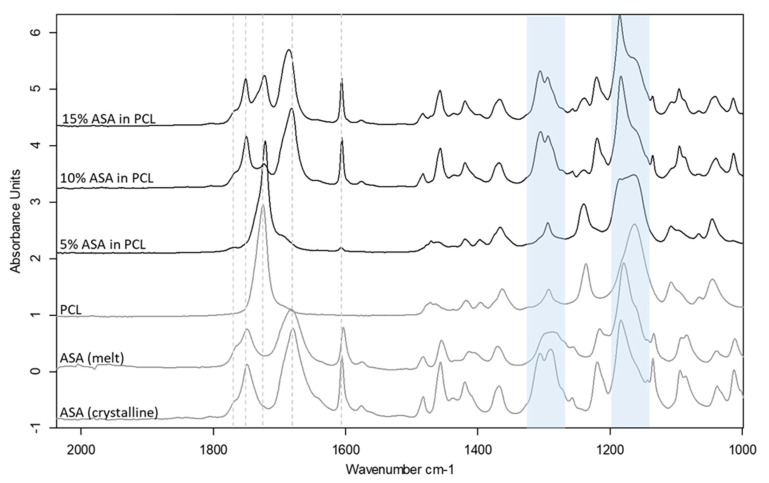
ATR-FTIR spectra of drug loaded PCL filaments in comparison to the raw material and the placebo filament. Shifted C-O stretching peak regions at ~1160 cm^−1^ and ~1290 cm^−1^ are highlighted in blue.

**Figure 7 pharmaceutics-11-00633-f007:**
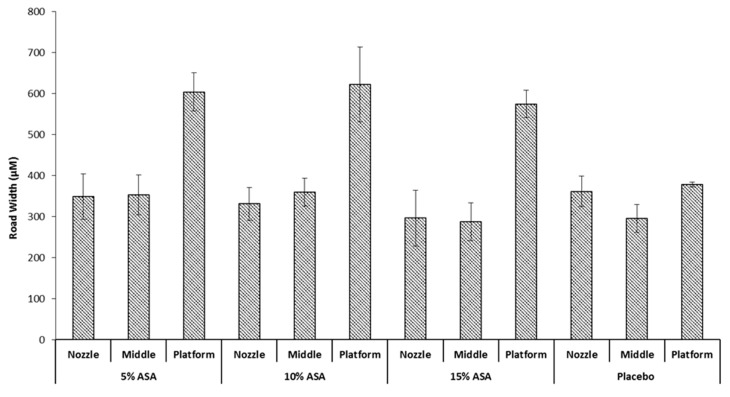
Impact of drug loading on the road width of the printed films.

**Figure 8 pharmaceutics-11-00633-f008:**
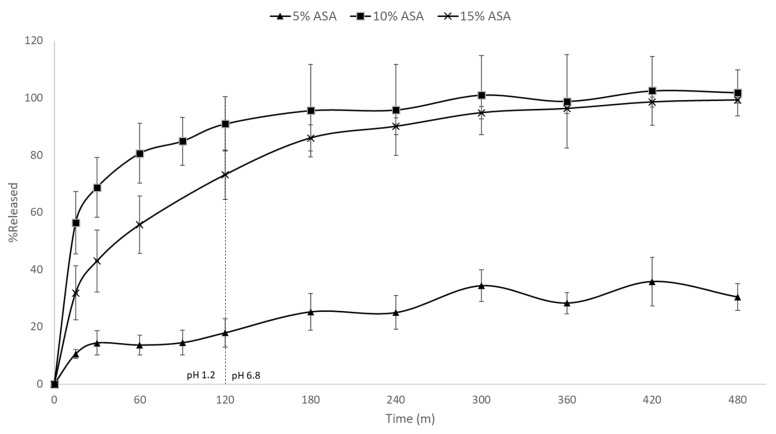
In vitro drug release profiles of FDM printed PCL films containing 5%, 10%, and 15% ASA.

**Figure 9 pharmaceutics-11-00633-f009:**
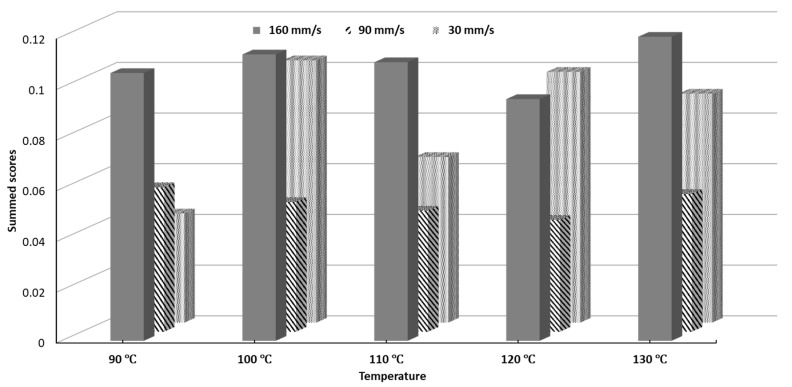
Summed Standard Deviation (SSD) scores of printability of all tested conditions.

**Figure 10 pharmaceutics-11-00633-f010:**
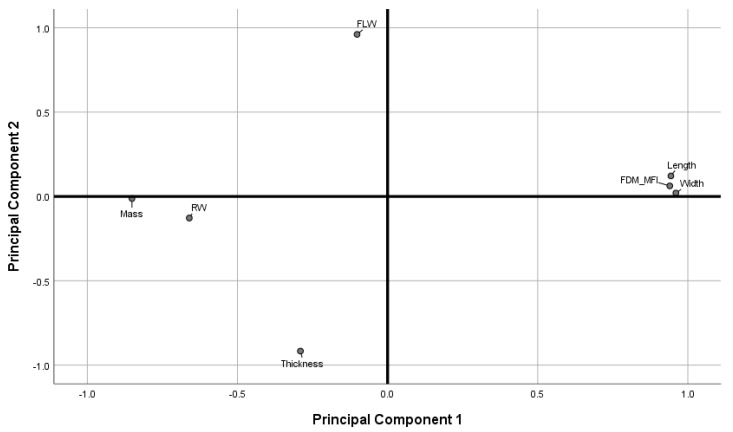
Loadings plot in rotated space. (FLW: first layer width. RW: road width).

**Figure 11 pharmaceutics-11-00633-f011:**
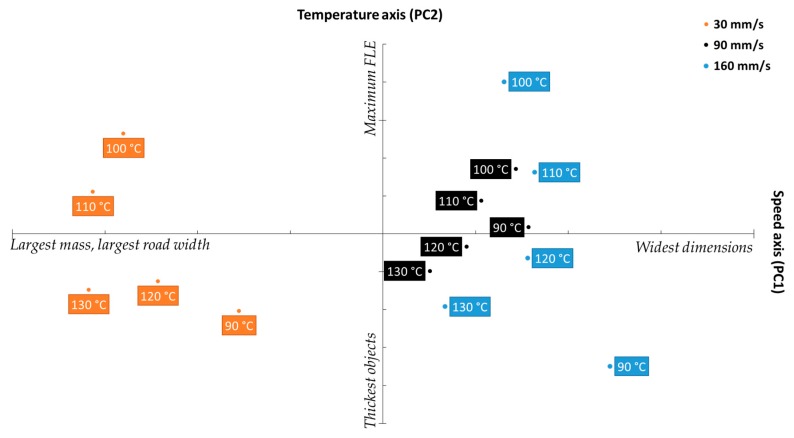
Biplot projecting the scores of the studied cases onto the response loadings.

**Table 1 pharmaceutics-11-00633-t001:** Experimental parameters used for printing the selected 3D object using MakerBot^®^ Flexible filament.

Nozzle Temperature (°C)	Platform Temperature (°C)	Printing Speed (mm/s)
100	30	30
100	30	90
100	30	160
110	30	30
110	30	90
110	30	160
120	30	30
120	30	90
120	30	160
100	45	30
100	45	90
100	45	160

**Table 2 pharmaceutics-11-00633-t002:** Effect of leveling by different operators on the weight and dimensions of the printed objects. RSD: Relative standard deviation ((Standard deviation/Mean)×100).

**Operator I**				
**n**	**Thickness (mm)**	**Length (mm)**	**Width (mm)**	**Weight (mg)**
1	1.09	9.92	10.01	110.6
2	1.04	10.03	10.10	114.6
3	1.07	10.00	10.16	113.4
4	1.05	10.00	10.09	111.0
5	1.10	9.91	10.02	109.8
6	1.05	9.99	10.20	111.0
RSD	2.25%	0.48%	0.74%	1.65%
**Operator II**				
**n**	**Thickness (mm)**	**Length (mm)**	**Width (mm)**	**Weight (mg)**
1	1.03	10.05	10.16	107.7
2	1.08	10.05	10.23	112.6
3	1.04	10.06	10.13	111.1
4	1.06	10.05	10.20	112.5
5	1.04	9.96	10.18	108.7
6	1.06	9.94	10.10	110.1
RSD	1.71%	0.53%	0.46%	1.8%
**Independent Sample *T*-Test**				
*p*-Value (Length)		*p*-Value (Width)	*p*-Value (Thickness)	*p*-Value (Weight)
0.223		0.101	0.127	0.137

**Table 3 pharmaceutics-11-00633-t003:** Inter-day variation in leveling on the weight and dimensions of the printed objects. RSD: Relative standard deviation ((Standard deviation/Mean)×100).

**Day 1**				
**n**	**Thickness (mm)**	**Length (mm)**	**Width (mm)**	**Weight (mg)**
1	1.09	9.92	10.01	110.6
2	1.04	10.03	10.10	114.6
3	1.07	10.00	10.16	113.4
4	1.05	10.00	10.09	111.0
5	1.10	9.91	10.02	109.8
6	1.05	9.99	10.20	111.0
RSD	2.25%	0.48%	0.74%	1.65%
**Day 2**				
**n**	**Thickness (mm)**	**Length (mm)**	**Width (mm)**	**Weight (mg)**
1	1.02	10.08	10.11	108.7
2	1.04	10.09	10.12	108.4
3	1.07	10.03	10.14	105.2
4	1.03	10.06	10.11	107.9
5	1.02	10.03	10.08	104.5
6	1.04	10.06	10.14	109.7
RSD	1.83%	0.25%	0.23%	1.93%
**Independent Sample *T*-Test**				
*p*-Value (Length)		*p*-Value (Width)	*p*-Value (Thickness)	*p*-Value (Weight)
0.004		0.213	0.04	0.006

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
