# Peer review of "Impact of Processing Parameters on the Quality of Pharmaceutical Solid Dosage Forms Produced by Fused Deposition Modeling (FDM)"

_pharmaceutics, 2019, doi:10.3390/pharmaceutics11120633_

Round 1
Reviewer 1 Report
This manuscript is described the findings of an investigation into a number of critical process parameters of FDM and their impact on quantifiable, pharmaceutically-relevant measures of quality.
It seems useful for pharmaceutical industry, but there are several weak points. This manuscript should be revised for publication.
#1: Line218 Please describe about the condition of “calibration” prints.
#2: Tabel 2. The author evaluated the effect of levelling by different operators on the several parameters of the printed objects. However, inter- and intra-day difference is also important. I suggest to author to evaluate inter and intra-day difference.
In addition, the author showed STDEV of each parameter. However, coefficient of variance is also important for understand of variation. Please show the coefficient of variation of each parameter.
#3: Fig2. The printable region of temperature is between 100 and 130oC. Is the material loaded on FDM 3D printing kept these temperatures? If the materials kept around 100oC, some drugs may degrade during print. Please discuss about stability of material loaded on FDM 3D printing machine.
Author Response
We appreciate the time and effort that you and the reviewers have dedicated to providing your valuable feedback on my manuscript. We are grateful to the reviewers for their insightful comments on our research. We have been able to incorporate changes to reflect most of the suggestions provided by the reviewers. We have highlighted the changes within the manuscript. Below is a point-by-point response to the reviewers’ comments and concerns:
Reviewer 1: 1) Line218 Please describe about the condition of “calibration” prints. - Thank you for pointing this out. We agree with this comment and the manuscript has been modified to reflect the recommended changes by detailing the calibration prints in the methods section (section 2.2.5, lines 178– 187).
2) Table 2. The author evaluated the effect of levelling by different operators on the several parameters of the printed objects. However, inter- and intra-day difference is also important. I suggest to author to evaluate inter and intra-day difference. In addition, the author showed STDEV of each parameter. However, coefficient of variance is also important for understand of variation. Please show the coefficient of variation of each parameter. - Thank you for this suggestion. We agree with the reviewer and performed the additional tests. The data are presented in Table 3 in the revised manuscript showing inter-day variation when the build plate was levelled by the same operator (Lines 258 – 259). Tables 2 and 3 have been updated to include the coefficient of variation (referred to as Relative Standard Deviation in-text) according the reviewer’s recommendation. Furthermore, the discussion and the conclusion sections were updated to incorporate the findings presented in Tables 2 and 3 (Line 245-267 and Line 562-566).
3) Fig2. The printable region of temperature is between 100 and 130 °C. Is the material loaded on FDM 3D printing kept these temperatures? If the materials kept around 100 °C, some drugs may degrade during print. Please discuss about stability of material loaded on FDM 3D printing machine. - We thank you for raising this point. Thermal stability of pharmaceutical products is an important factor and mitigating against degradation is important in melt polymer processing. Regarding the data presented herein, the material loaded in the printer was not held isothermal at that temperature, the mechanism of printing is the material displacement in the nozzle. During the printing, the filament was kept cold and was driven into the melting zone by the rollers when required, the exposure of the material to the printing temperature was only for a few seconds at most when the material was fed into the printing nozzle. We also performed TGA tests on the drug alone and the filaments to check the degradation temperature of the drug. The results confirmed that at 130 °C no drug degradation occurred.
Reviewer 2 Report
Dear Authors,
1) You performed an interesting manuscript about the "Impact of processing parameters on the quality of pharmaceutical solid dosage forms produced by fused deposition modelling (FDM)".
2) Several quality attributes of the 3D printed dosage forms, such as weight, dimensional authenticity, road width, and overall print reproducibility, were studied. However, in my opinion, the evaluation of the mechanical strength (hardness and friability) as well as the drug release characteristics (dissolution rate and disintegration time) should be studied. Please comment in detail on these aspects.
3) Images of the 3D printed dosage forms (macroscopic or obtained by scanning electron microscopy) should be included.
4) In order to enrich the article, the fused deposition modelling (FDM) process should be compared (advantages/disadvantages/differences) with other 3D printing processes (for instance, semisolid extrusion 3D printing) in the printlets development. In addition, recent manuscripts should be added and discussed.
Author Response
We appreciate the time and effort that you and the reviewers have dedicated to providing your valuable feedback on my manuscript. We are grateful to the reviewers for their insightful comments on our research. We have been able to incorporate changes to reflect most of the suggestions provided by the reviewers. We have highlighted the changes within the manuscript. Below is a point-by-point response to the reviewers’ comments and concerns:
Reviewer 2:
1) You performed an interesting manuscript about the "Impact of processing parameters on the quality of pharmaceutical solid dosage forms produced by fused deposition modelling (FDM)". Several quality attributes of the 3D printed dosage forms, such as weight, dimensional authenticity, road width, and overall print reproducibility, were studied. However, in my opinion, the evaluation of the mechanical strength (hardness and friability) as well as the drug release characteristics (dissolution rate and disintegration time) should be studied. Please comment in detail on these aspects. - We thank the reviewer for the comment. We agree with and acknowledge the importance of the routinely performed quality control tests (i.e. hardness and friability). However, in our case the hardness and friability, which are key quality attributes of conventional tablets, are not applicable to the FDM printed dosage forms produced in this study. In this study, we printed 3-layered square-shaped oral films. - Regarding the dissolution rate and disintegration time. We agree with this comment and acknowledge the significance of drug release characterisation. Therefore we have added in the in vitro drug release section in both methods (Line 154-160) and results and discussion sections (Line 432-448). In section 3.7 (in vitro drug release) we describe the characterisation of release properties of the three drug-loaded formulations presented in the manuscript. As PCL is a biodegradable polymer, no disintegration was observed within the 8 hours of in vitro dissolution tests.
2) Images of the 3D printed dosage forms (macroscopic or obtained by scanning electron microscopy) should be included. - Thank you for bringing this comment to our attention. Macroscopic and microscopic images of the 3D printed dosage form have been added as the new Fig. 1 to the main manuscript in the section 2.2.2 (line 142-143).
3) In order to enrich the article, the fused deposition modelling (FDM) process should be compared (advantages/disadvantages/differences) with other 3D printing processes (for instance, semisolid extrusion 3D printing) in the printlets development. In addition, recent manuscripts should be added and discussed. - Thank you for this comment and recommendation. The text of the manuscript has been revised and modified to provide wider review of the recent developments of 3D printing in solid dosage forms in the introduction section (Line 39-60).
Round 2
Reviewer 1 Report
This manuscript is well revised and it is suitable for publication.
Reviewer 2 Report
Dear Authors,
With the changes performed, the article has improved considerably. Thus, I consider that the manuscript can be published.